# Spatio-Temporal Variations of Volatile Metabolites as an Eco-Physiological Response of a Native Species in the Tropical Forest

**DOI:** 10.3390/plants13182599

**Published:** 2024-09-18

**Authors:** Jéssica Sales Felisberto, Daniel B. Machado, Jeferson A. S. Assunção, Samik A. S. Massau, George A. de Queiroz, Elsie F. Guimarães, Ygor J. Ramos, Davyson de Lima Moreira

**Affiliations:** 1Postgraduate Program in Plant Biology, State University of Rio de Janeiro, Maracanã, Rio de Janeiro 20550-013, RJ, Brazil; jessicka.salles@gmail.com (J.S.F.); danielmachado.ceade@gmail.com (D.B.M.); or ygorjesse@jbrj.gov.br (Y.J.R.); 2Natural Products and Biochemistry Laboratory, Rio de Janeiro Botanical Garden Research Institute, Jardim Botânico, Rio de Janeiro 22460-030, RJ, Brazil; samiklourencomassau@hotmail.com (S.A.S.M.); elsie.guimaraes@jbrj.gov.br (E.F.G.); 3Earth’s Pharmacy Laboratory, Federal University of Bahia, Ondina, Salvador 40170-215, BA, Brazil; 4Postgraduate Program in Translational Research in Drugs and Medicines, Pharmaceutical Technology Institute (Farmanguinhos), Oswaldo Cruz Foundation, Rio de Janeiro 21041-250, RJ, Brazil; jeersonadriano.sa@gmail.com; 5Department of Pharmacy, State University of Rio de Janeiro, Rio de Janeiro 23070-200, RJ, Brazil; georgeazevedo08@gmail.com

**Keywords:** chemodiversity, chemophenetic, essential oils, *Piper rivinoides*, Piperaceae, metabolic signature

## Abstract

This study evaluates the essential oil (EO) composition of *Piper rivinoides* Kunth, a shrub native to the Brazilian tropical rainforest, across different plant parts and developmental phases. The aim was to explore the chemical diversity of EO and its reflection in the plant’s ecological interactions and adaptations. Plant organs (roots, stems, branches, and leaves) at different developmental phases were subjected to hydrodistillation followed by chemical analysis using Gas Chromatography–Mass Spectrometry (GC–MS) and Gas Chromatography–Flame Ionization Detector (GC–FID). The results revealed a relevant variation in EO yield and composition among different plant parts and developmental phases. Leaves showed the highest yield and chemical diversity, with α-pinene and β-pinene as major constituents, while roots and stems were characterized by a predominance of arylpropanoids, particularly apiol. The chemical diversity in leaves increased with plant maturity, indicating a dynamic adaptation to environmental interactions. The study underscores the importance of considering the ontogeny of plant parts in understanding the ecological roles and potential applications of *P. rivinoides* in medicine and agriculture. The findings contribute to the overall knowledge of Piperaceae chemodiversity and ecological adaptations, offering insights into the plant’s interaction with its environment and its potential uses based on chemical composition.

## 1. Introduction

Special metabolites in plants reflect their needs and interactions with the environment, with differences in metabolic pathways between organs as an eco-physiological response. The synthesis, allocation, and distribution of these metabolites are dynamic and complex processes that not only mirror the plant’s genetics, but also its interactions with its geographical origin and environmental conditions [1,2]. These physiological responses involve intracellular signaling pathways, including plant hormones, gene expression regulation, protein kinase activation, and secondary metabolite synthesis [3]. This synthesis is a critical aspect of ecophysiology, illustrating how plants chemically react to environmental factors, forming a unique metabolic signature [4].

Variations in the structural composition of plant organs can lead to differences in primary metabolic pathways as a response to ecological and physiological factors, reflecting the strong link between structure and function [1,5]. Additionally, the chemical traits of plants can be influenced by spatial and temporal changes, and metabolic functions may undergo significant alterations during developmental stages [6,7,8]. Rapid changes in chemical phenotypes can influence a plant’s immediate environment, shaping its ecological niche. Plants adapt their chemical responses and phenotypes to better align with environmental conditions, enhancing fitness in specialized ecological niches [9,10,11,12]. However, it remains unclear if this adaptive plasticity in chemical traits is also evident among wild plant populations in natural ecosystems.

Chemical diversity within natural ecological niches is a crucial tool for studying ecophysiology [9,10,11]. Examining chemical interactions within different plant parts, their eco-physiological acclimation in tropical environments, and phenotypic changes during development enriches our understanding of plant chemical ecology and adaptive capacities [13]. These insights could be valuable for biodiversity conservation, understanding plant–environment interactions, and exploring the potential applications of secondary metabolites in agriculture, industry, and medicine.

This study focuses on *Piper rivinoides* Kunth (Piperaceae), a shrub native to the Brazilian tropical rainforest, commonly found in shaded areas near trails, riverbanks, and wetlands [14,15]. *P. rivinoides* plays a significant role in ecological resilience and ecosystem recovery, with its leaves extensively used in Brazil for medicinal and ritual purposes, including wound healing, ulcers, vaginal discharge, bleeding, and oral health issues [16,17,18,19,20,21,22,23]. Despite its uses, there is no commercially cultivated variety, with people relying instead on wild specimens. Recent phytochemical studies reveal a diverse range of secondary metabolites, such as neolignans, terpenes, and arylpropanoids, highlighting their potential for bioactivity [18,19,24,25,26]. This study aimed (1) to understand how resource allocation occurs in a spatio-temporal context in the production of secondary metabolites, particularly essential oils (EO); and (2) to investigate the chemical diversity of EO obtained at different organ and ontogeny levels using chemodiversity and advanced chemophenetic metrics.

## 2. Material and Methods

### 2.1. Botanical Material

*P. rivinoides* (roots, stems, branches, and leaves) were collected in January during the summer of 2022 at 9:00 a.m. (authorization No. 07/0002.007362/2021) in the Pedra Branca State Park, in the city of Rio de Janeiro, Brazil (22°58′12″ S, 43°14′30″ W, altitude 452 m).The Pedra Branca State Park is located in the city’s Western zone and is recognized as the world’s largest urban forest, with a humid tropical climate and no dry season [27]. For the study of plant development, only the leaves were collected to maintain species conservation and minimize environmental impact. As there were no botanical studies on the developmental stages of the *P. rivinoides* plant, we used morphological characteristics such as branching degree and plant height above the ground to define these stages. Thus, we established the following developmental stages: phase I—25 cm tall with an unbranched herbaceous stem; phase II—40 cm tall with a branched herbaceous stem; phase III—70 cm tall with a herbaceous stem with 3 or more branches; phase IV—2 m tall with a lignified stem and multiple branches; phase V—7 m tall with a thick stem, multiple branches, and lignified [28,29]. During the collections, the third pair of leaves (from top to bottom), corresponding to the first well-expanded leaf, was sampled. The collected leaves were mature, without signs of cloning, herbivore damage, or reproductive organs. To ensure sample representativeness, specimens were selected whenever at least five plants of similar size were found close to each other, serving as replicas. All collections were made within a 30 m radius to ensure genetic uniformity. Dr. George Queiroz de Azevedo from the Research Institute of the Rio de Janeiro Botanical Garden conducted the botanical identification of the plant material, and herbarium samples were deposited at the Herbarium of Botanical Garden of Rio de Janeiro (RB) under voucher numbers (RB 861754). This study was registered in the National System for the Management of Genetic Heritage and Traditional Knowledge (SISGEN) under number AE4E953.

### 2.2. Essential Oil Extraction and Analysis

The different plant organs at different phases of *P. rivinoides* (100 g each) were separately subjected to hydrodistillation using a modified Clevenger-type apparatus for two hours. The resulting EO was separated from the aqueous phase, dried with anhydrous sodium sulfate, and stored in dark amber flasks in a freezer at −20 °C until analysis [30]. The total EO yield was expressed as a percentage value, calculated as weight of EO (g) divided by the weight (g) of fresh plant × 100.

The obtained EOs were diluted in dichloromethane (1 mg/mL) (Tedia, Brazil) and then subjected to Gas Chromatography–Mass Spectrometry (GC–MS) analysis for identification purposes and Gas Chromatography–Flame Ionization Detector (GC–FID) for quantification analysis [31]. The GC–MS conditions used were as follows: analysis was carried out using an HP Agilent GC 6890 gas chromatograph coupled to an Agilent MS 5973N mass spectrometer (Santa Clara, CA, USA), with an ionization energy of 70 eV (positive mode). The EO solution was injected at 1 μL (splitless) and the injector temperature was set at 270 °C. The sample was run through an HP-5MS capillary column (30 m × 0.25 mm i.d. × 0.25 μm film thickness) (Agilent J&W, Santa Clara, CA, USA), with an oven temperature program ranging from 60 °C to 240 °C, with an increase of 3 °C/min (60 min total run), using helium (>99.99%) as the carrier gas, at a constant flow rate of 1 mL/min. The monitoring mass range was *m*/*z* 40–600 atomic mass unit (*u*). The GC–FID analysis was carried out using an HP-Agilent 6890 GC–FID gas chromatograph in the same conditions as for GC–MS, except using hydrogen as the carrier gas at a constant flow rate of 1.0 mL/min. The retention times (tR) were measured in minutes without correction [30,31]. Retention index (RI) and the peak area quantification were obtained based on the GC–FID results. The relative percentage of individual components was calculated based on GC peak areas without FID response factor correction. Linear retention indices (RIs) were calculated for separate compounds relative to *n*-alkanes (C_8_–C_28_, Sigma-Aldrich, São Paulo, SP, Brazil). Constituents were identified by comparison of their calculated RIs with those in the literature [32], and by comparison of the mass spectrum with those recorded by the NIST library (National Institute of Standards and Technology) “NIST14” and Wiley (ChemStation data system) “WILEY7n” [33]. Additionally, authentic pattern co-injection was performed whenever possible [33].

### 2.3. Data Processing and Statistical Analysis

In this study, data are presented as the mean ± standard deviation, derived from triplicate analyses, to ensure reliability of results. To investigate the relationships and variations within plant organs, were applied different statistical methods to our dataset. The results from GC–MS were treated as operational taxonomic units (OTUs) and transformed using the arcsine square root method. These data were then converted into a matrix using percentage area (%), excluding unidentified substances or those with less than 1% content [34]. Subsequently, Principal Component Analysis (PCA) and Hierarchical Cluster Analysis (HCA) was applied to examine similarities among OTUs, based on the class and distribution of chemical compounds in different organs and phases. For dendrogram generation, we used Euclidean distances with the Unweighted Pair Group Method with Arithmetic Mean (UPGMA) [34]. Additionally, the following analyses were conducted: I—variations between the chemical compositions of organs; II—chemical diversity among different compartments; III—variations in micromolecular evaluation indices and diversity. To compare the obtained means, analysis of variance (ANOVA) was employed using Statistica software, version 13 (StartSoft Inc., Tulsa, OK, USA). Means were subjected to Tukey’s test at a significance level of 5% probability [35].

### 2.4. Chemodiversity and Advanced Chemophenetics Approaches

To measure the chemodiversity among different organizational and spatial levels of the plant, the content of the chemical composition of the EO (percentage value % of the area), extracted from data obtained by GC–FID, were treated as operational taxonomic units (OTUs) and subjected to calculations of (1) *Shannon index* (*H*’); (2) *Pielou index* (*E*) for chemodiversity α; (3) *Jaccard index* (C_J_); and (4) *Sorensen Index* (SI) for chemodiversity β. In relation to advanced chemophenetic analysis, the (5) *Weighted Oxidation-Reduction Level* (N_OR_) and (6) *Ramos and Moreira* Index (EI_R&M_ R&M index) were calculated [6,36,37,38] The equations for these indices are provided below.
(1)$$H’=-\sum P_{i}lnP_{i}$$
(2)$$E=\frac{H’}{S}$$
(3)$$C_{J}=\frac{C}{A+B-C}$$
(4)$$SI=\frac{2C}{A+B}$$
(5)$$N_{\mathrm{OR}}=\frac{\left(N_{\mathrm{ox}}\times Q\%\right)}{n}$$
(6)$${\mathrm{EI}}_{R\&M}=\frac{\sum N_{\mathrm{OR}}}{N_{\mathrm{SI}}}$$

In these equations: (1) *Pi* represents the proportional abundance of each compound, calculated by dividing the amount of the identified compound by the total amount of all identified compounds in the sample, where the total number of compounds are present in the sample. (2) *H*’ is the value of the Shannon index, and *S* is the number of compounds; (3,4) A corresponds to the number of compounds in one sample; B is the number of compounds in another sample, and C is the number of the same compounds found in both A and B. (5) Values for N_OR_ were obtained by the Nox of the compound, which is the oxidation number of the chemical structure, Q% is the relative content obtained for each compound from the GC–FID, and *n* is the number of carbon atoms in the chemical structure. (6) Values for IE_R&M_ were obtained by the ratio of the sum of all N_OR_ values for each compound to the number of identified compounds in the sample (N_SI_).

## 3. Results

### 3.1. Chemical Composition and Yields of Essential Oils from P. rivinoides

Information about the chemical composition, yield, and number of compounds identified in the EO from different plant parts (>0.2%) is recorded in Table 1. The EO obtained from the leaves showed the highest yield (0.82%, *w*/*w*), followed by the roots (0.57% *w*/*w*), branches (0.31% *w*/*w*), and stem (0.05% *w*/*w*). A total of 111 compounds were identified in the EO. The chemical composition of leaf and branch was quite similar, with α-pinene (1) (31.90% in leaves and 20.87% in branches) and β-pinene (2) (20.96% in leaves and 64.61% in branches) being identified as the major constituents. The compound present in the highest amount in root and stem was apiole (3), with relative percentages of 69.68% and 59.32%, respectively (Figure 1). The leaves and roots exhibited the highest numbers of identified compounds, with 30 and 24, respectively, and 11 compounds in common (Figure 2b). The compound α-pinene was identified in all plant organs.

The PCA revealed the formation of two clusters with a total variance of 90.09%. Cluster I, comprising roots and stems, was characterized by EO with a high content of arylpropanoids, primarily represented by apiol (−11.54 PC1). Cluster II, consisting of leaves and branches, grouped EO rich in monoterpenes, represented by α-pinene (1) (+6.83 PC2) and β-pinene (2) (+9.03 PC1) (Figure 3a). The HCA of the EO obtained from different plant organs showed the formation of two clusters, reflecting mainly the relative percentage content of arylpropanoids in Cluster I and monoterpenes in Cluster II (Figure 3b).

Details about the chemical composition, yield, and number of compounds identified in the ontogeny study (compounds with >0.2% EO) are provided in Table 2. A total of 95 compounds were successfully identified, including monoterpenes, sesquiterpenes, and arylpropanoids. The Venn diagram analysis (Figure 2a) revealed that eight compounds (α-pinene, β-pinene, camphene, carene, myrcene, *E*-caryophyllene, cymene, and limonene) are present at all stages of ontogeny. To access all the data from Venn diagram, see Appendix A. Furthermore, as the plant size increased, there was a higher diversification and synthesis of novel chemicals from mevalonate, methylerythrose phosphate, and shikimate pathways. Specimens at phase IV and V exhibited the highest number of shared compounds, totaling 18 (Figure 2a). Specimens at phase I and II showed chemical similarities regarding the arylpropanoids apiole (59.59–74.69%) and dillapiole (2.29–2.76%) as major constituents. Specimens at phase III, IV, and V predominantly displayed non-oxygenated monoterpenes (α-pinene: 20.03–35.09%; β-pinene: 11.45–16.72%; and δ-2-Carene 0.32–42.07%). PCA analysis revealed the formation of two clusters, driven by the antagonistic influences of two compounds: group one included phase I and II (rich in apiole), while group two encompassed phases III, IV, and V (rich in α-pinene) (Figure 4a). HCA also confirmed this ontogenetic distinction based on the biosynthetic pathway (Figure 4b).

### 3.2. Chemodiversity and Chemophenetic Index

All calculations of chemodiversity α and β consider the same data matrix. For the EO obtained from different organs, it was possible to observe that the Shannon index, which assesses compound diversity (richness), showed the highest value in leaves (2.45), followed by roots (1.40), branches (1.26), and stems (0.93). Conversely, the Pielou index assesses the uniformity of the volatile mixture (evenness) and indicates how the number of compounds is distributed within it. The uniformity index provides a projection of the maximum diversity of the mixture, and once again, leaves exhibit the highest uniformity value among the relative percentage of compounds (0.62), followed by stems (0.48), branches (0.38), and roots (0.35) (Figure 5) [6,36,37,38].

Regarding ontogeny, younger plants (phase I and phase II) showed lower compound diversity (with a Shannon index of 1.12 and 1.49, respectively). Conversely, adult plants registered an increase in diversity, reaching the highest values among plants measuring phase V (2.58) and III (2.56), followed by those at IV (1.96). The Pielou index in these volatile mixtures demonstrated that the uniformity in plants with III was the highest value among the relative percentage of compounds (0.72), followed by plants in V (0.64), IV (0.49), II (0.45), and the lowest value was found in young plants of I (0.36) (Figure 5).

Chemodiversity β indices are commonly used to compare spatial similarities between different specimens. However, it is possible to apply these indices by treating each organ as an independent region or “space” of the plant [6]. The Jaccard index indicates the proportion of shared compounds between samples relative to the total number of compounds in each plant organ. In general, samples of different organs showed low chemical similarity. Nevertheless, the highest similarity was noted between branches and leaves (Jaccard index of 31.00), as well as between roots and leaves (Jaccard index of 28.57). The Sorensen index also represents similarity between samples, assigning greater weight to the presence of compounds in the sample than to the absence data of these. The Sorensen index indicated a greater resemblance between roots and leaves (44.00). However, both β indices showed low chemical similarity among the different organs (Table 3).

In relation to the different ontogenic phases, the highest similarity values, both for the Jaccard index (75.00) and the Sorensen index (86.00), were found between the samples in IV and V phase. Conversely, the most dissimilar samples were those of I compared to III, with values of 3.85 for the Jaccard Index and 7.00 for Sorensen (Table 3).

The values of the R&M index were negative for all plant organs. They ranged from −2.01 in roots to −14.71 for stems. This indicates that the mixtures of volatile compounds in the EO in all plant studied compartments are reduced. This can be a particularity of the plant or suggest some physiological trend. The sum of the highest N_OR_ values was found in branches (−158.43), followed by leaves (−155.42), roots (−98.30), and stems (−88.35).

Considering ontogeny, the R&M index ranged from 3.09 to 3.93, that is, it also presented negative values, albeit with less variation than for the different organs.

## 4. Discussion

### 4.1. Chemical Composition and Yields of Essential Oils from P. rivinoides

The plant organs that exhibited higher yields of EO were roots and leaves. This result was consistently observed in various plant families, including Piperaceae, Asteraceae, Lamiaceae, and Myrtaceae [39,40,41,42]. Consequently, leaves and roots tend to contain a higher content of volatile compounds, compared to branches and stems [43].

In nature, plants face a wide range of biotic and abiotic factors, and this evolutionary pressure stimulates the development of adaptation mechanisms to the environment. In this context, EO production is a biological response resulting from complex and nuanced interactions between genotype and environment, involving changes in the expression of one or more genes in response to various environmental stimuli, such as UV radiation, temperature variations, microbial attacks, and herbivory, among others [44,45,46]. Plant leaves are particularly susceptible to these environmental stresses, as they play a fundamental role in photosynthesis and gas exchange processes. Therefore, they develop defense mechanisms against these stressors. Conversely, roots, responsible for water and nutrient absorption, maintain intimate contact with the soil and its microbiome, which include many pathogenic fungi and bacteria [47]. Consequently, these organs, compared to stems and branches, are subjected to a greater diversity of environmental stimuli. As a result, the production of high content of EO emerges as an adjustment between genotype, environment, and phenotypic expression, aiming to promote an increase in plant fitness, favoring survival and reproduction in challenging environments [3,48].

Another important factor to consider in these variations is the anatomical structures that store EO, as well as the ease of extraction and the cellular organization of plant tissue. This includes elements such as the number, shape, size, distribution, location, and density of these structures [49]. For example, certain anatomical structures may provide greater accessibility to volatile compounds during the extraction process, such as oil glands present in leaves, which may be more readily exposed water vapor during extraction, resulting in a higher yield of EO. Additionally, leaf cells may be more prone to release EO due to their delicate cellular structure compared to stem or root cells, which may be denser and more resistant due to high lignification. Therefore, differences in plant anatomical structures may have a significant impact on the yield of extracted EO, both in terms of quantity and quality [50].

In the study of ontogeny, variations in the yields of EO do not allow us to make conclusive inferences due to the oscillation and lack of trend in the results. The absence of a pattern may be attributed to the existing natural variation in the plant. It is important to recognize that EO production in plants is a complex and multifaceted process, involving interactions between a variety of biotic and abiotic factors, as mentioned. Therefore, in some specific cases, it may be challenging to detect clear patterns in field studies due to the complexity of the system.

Regarding the chemical composition, the major constituents found in the samples were the monoterpenes α-pinene, β-pinene, and the arylpropanoid apiole (Figure 1). The Venn diagram showed that the monoterpene α-pinene is the only compound present in all organs of *P. rivinoides*, and in all phases of the plant’s ontogenetic development (Figure 2). It is important to note that α-pinene is a common monoterpene found in the EO of coniferous trees, such as the genera *Pinus* and *Picea*, as well as in the genus *Eucalyptus* [51,52,53]. α-Pinene has two structural isomers, α-pinene and β-pinene, which are responsible for the characteristic aromas of pine and turpentine oil, respectively. Additionally, two pairs of enantiomeric isomers of these monoterpenes naturally coexist in nature: (−)-α-pinene and (+)-α-pinene, and (−)-β-pinene and (+)-β-pinene [54,55]. However, in our analysis, it is not possible the separation of the enantiomers (−) and (+).

PCA showed greater similarity among the chemical constituents of leaves and branches compared to roots and stems, although the Euclidean distance between clusters was relatively low. These differentiation trends among Piperaceae organs have been demonstrated in the literature [6,56]. The biosynthesis of α-pinene and β-pinene occurs via the mevalonate pathway from geranyl pyrophosphate (GPP) through the cyclization of linalyl pyrophosphate, followed by proton loss. Consequently, their biosynthesis is considered less complex, occurring at the beginning of the terpene biosynthetic pathway [57]. Although studies on the chemical/ecological function and functionality of pinene isomers are incipient, they encompass a wide range of functions. Actions include protection against herbivores [58]; repellent action [51]; antibacterial activity [59,60]; attraction of pollinators with insect–plant interaction [61], where they perform signaling functions within a population/community or even in adjacent plants; an inhibitory effect on the spore formation of *Metarhizium* fungi, even at concentrations below 5% [62]; regulation of leaf temperature [62,63,64]; participation in allelochemical communication [65]; inhibition of root growth [65,66]; induction of oxidative stress [67,68]; alteration of chlorophyll content; and decrease in seedling biomass [68]. Overall, the significant presence of pinenes, especially α-pinene, becomes more prominent in leaves and branches compared to stems and roots. It has been demonstrated that pinenes play a role in conifer resistance against beetles and associated fungi [55]. Therefore, it is plausible that *P. rivinoides* accumulates higher content of these pinenes in its leaves as a physiological response to external stimuli. The monoterpene cyclase enzyme has been reported to increase its activity (U) in response to elicitors, which are often found in insect saliva. This suggests that the production of α-pinene and β-pinene may function as an inducible chemical defense, particularly in leaves, though roots also show a response [20,69].

Another important point to consider is that *P. rivinoides* is probably consumed by generalist herbivores, as different patterns of herbivory have been observed in the field on the leaves (unpublished data), suggesting that this organ is attacked by different insects. If a plant’s defenses are specifically adapted to combat a particular herbivore and require a high investment of resources, there may be trade-offs between different strategies. Conversely, if defenses are nonspecific, the same mechanism can increase the plant’s ability to resist different aggressors, leading to synergistic or positive associations. In this sense, extrinsic factors of the plant may cause a lower energetic investment in the production of more complex and specific defense structures and, instead, direct its energy reserves towards the synthesis of less complex compounds that require less energy expenditure [70], a hypothesis to be confirmed.

However, it is important to emphasize that this research addresses only a specific moment of the eco-physiology of *P. rivinoides*. Previous studies on this plant report low seasonal variation in the composition of EO, indicating that the predominant biosynthesis of the monoterpenes α- and β-pinene occurs continuously throughout the year [19,71]. This absence of significant seasonal phenotypic variation is called seasonal monophenism [3]. Therefore, it is more likely that this continuous biosynthesis mechanism of pinene isomers at high content is considered an ideal defense mechanism, rather than an induced defense mechanism [20,69,72]. The theory of ideal defense suggests that defense compounds are produced or expressed in tissues with a higher likelihood or risk of being attacked [69,71,73]. Therefore, it is plausible that leaves and branches, being the structures responsible for photosynthesis and thus nutritional sources, accumulate a higher content of monoterpenes with allelopathic function in their tissues as an ideal defense mechanism against herbivory [69,71,74].

While monoterpenes were predominant in leaves and branches, in stems and roots the major component was the arylpropanoid apiol. Based on the theory of optimal defense and considering the capacity of the same genotype to express different phenotypes, influenced by the environment, it can be concluded that different defense and tolerance strategies in different organs of the same plant are independent of each other and occur in an optimized manner [3,75]. Thus, the presence of arylpropanoids in roots, stems, and young plants can be considered ideal defense mechanisms, as the synthesis of arylpropanoids promotes tissue production and, consequently, contributes to protection against pathogens [69,71]. It is important to note that stems and roots have constitutive defenses that ensure greater expression of lignification in the cell walls of these tissues, making them more resistant to mechanical damage. In addition, resource allocation to produce defensive compounds depends on the intensity of herbivore attacks, and different defense and tolerance strategies are independent of each other [3,76,77]. Therefore, it is suggested that the presence of a high content of arylpropanoids and low content of α-pinene in roots may also result from crosstalk with other organs, as roots are highly lignified underground organs [56,74,78]. Factors crosstalk integrate different physiological responses and optimizes the plant’s fitness based on the resources available in the organ in question.

Factors (genes, active enzymes, and hormones) of the organ in question dominate biosynthetic conductance. In other words, the accumulation of secondary metabolites in a specific organ comes from the production processes of the macromolecule related to the essential precursor [73,75,79]. For example, the roots of the studied plant accumulate a high content of arylpropanoids, compounds derived from the shikimate pathway, the same biosynthetic pathway for the formation of lignins (structural macromolecules). The roots and stems of *P. rivinoides* are highly lignified, which indicates that the shikimate pathway is more highly expressed in these organs. Therefore, there will be a greater accumulation of arylpropanoids, biosynthetic precursors of lignans and lignins, in these organs. Thus, considering that the metabolic pathway with the highest activity in this organ is the shikimate pathway, the biosynthesis of arylpropanoids may be energetically more economical for the roots. Additionally, the chemical phenotypic expression in roots tends to be strongly influenced by the soil microbiome, which can affect the ecological interactions between roots and pathogens [3].

During the different phases of plant ontogenetic development, relevant variations in major compounds were observed. In phases I and II, the major compound was the arylpropanoid apiol, while in phases IV and V, the monoterpenes α- and β-pinene predominated. This behavior can be interpreted as niche conformity, which is the process by which an individual adjusts its phenotype to the environment to better match it, thereby improving its fitness [10]. In a complex ecosystem like the forest, it is important to understand that the choice of niche in which the plant will develop does not occur actively as in the animal kingdom. Thus, the plant will only establish and thrive if the environmental conditions match the needs of the seedling [10,11,80]. Therefore, niche conformity emerges as a competitive advantage for the plant.

In other words, during the initial phase, the plant faces different adversities, such as low light incidence, allelopathic interactions with other plants attempting to inhibit their growth, and differential interaction with the fauna of the lower layers of the forest, among others [10,81]. The accumulation of arylpropanoids in plants during the initial phases of development has been observed in cultivated species of *Piper*, with this chemical class often associated with defense against herbivores [9,82]. It is important to consider that herbivory causes leaf damage that impairs photosynthesis, making it crucial to maintain the largest photosynthetically active leaf area possible to ensure the production of nutrients involved in plant growth. Conversely, when the plant is already established in the environment (phases IV and V), a majority accumulation of pinenes is observed. This possibly occurs because, at these stages, there is less of a need for the storage of compounds from the arylpropanoid class, because leaf damage caused by herbivory is no longer a significant problem due to the size of the canopy, height, and quantity of leaves of the plant. In addition, the lignification of the plant took place in high rates in the early stages of development allowing for the plant’s growth.

These ideas, along with the characteristics of the plant under study (a perennial plant that can reach up to 7 m height, with expanded leaves), lead to the inference that the modulation of biosynthesis and accumulation of chemical compounds in the leaves throughout plant development result from interaction with herbivores as well as growth rate, and can be understood as a “trade-off”. This term refers to the compensation between resource allocation of the plant for different objectives [68]. In this specific case, there is a direction of available energy and nutrients towards plant growth, sacrificing part of its leaf area lost during herbivory due to the absence of a more complex chemical defense. Thus, even with herbivory intensity, there is a remarkable growth of biomass (leaf area) of the plant. This compensation directly affects the reproductive success and survival of the plant.

In general, it is possible to extrapolate this biosynthetic modulation throughout plant development as an “Ontogenetic Niche Shift” or “Ontogenetic Chemical Niche Transition”, meaning that changes in the chemical profile of the plant reflect adaptations to new environmental conditions or biotic interactions [83,84,85]. Although these concepts are applied only in the field of ecology, because the study of the interaction between the plant and the environment also encompasses ecology, it is possible to transpose these concepts to variations in plant metabolites.

*P. rivinoides* stands out for its variety of secondary metabolites like terpenes, and arylpropanoids, which are not just byproducts but active agents in plant-environment interactions. These compounds function as hormone-like regulators and precursors to primary metabolites, crucial for the plant’s defense against environmental stressors and herbivores, thus enhancing its resilience and adaptability [79,86]. The biosynthesis of these metabolites may be influenced by both environmental and genetic factors, leading to variations in their production, such as in the EO under different environmental conditions [76].

Furthermore, the chemical diversity of these metabolites plays a significant role in shaping ecological interactions and community dynamics, with variations influencing the structure of herbivore and other organism communities interacting with *P. rivinoides* [87]. The organ-specific composition of these metabolites underscores their ecological and medicinal importance. This, coupled with the plant’s ability to rapidly alter its chemical phenotype in response to the environment, highlights *P. rivinoides*’ vital role in ecosystems and its evolutionary adaptability to varying conditions [88,89].

### 4.2. Chemodiversity and Chemophenetic Indices

The results of the calculations of chemodiversity α indices showed that the richness of chemical compounds present in the leaves of *P. rivinoides* is almost twice the richness found in the roots, which was the second organ with the highest chemical diversity. Given that leaves are the organs with the most significant interspecific interaction in a plant, it is postulated, for example, that the chemical diversity of leaves correlates with the diversity of the insect community interacting with them. Richards et al. (2015) sought to comprehend the role of plant chemodiversity in the intricate dynamics of ecological relationships within a community. They concluded that plants producing a greater variety of chemical compounds interact with a more diverse insect community. This factor may result in the different patterns of herbivory observed in the leaves of *P. rivinoides* [75].

Chemodiversity is also increased as the plant grows. There is also a significant correlation between developmental stages and the composition of EO. Smaller plants may have access to limited resources such as nutrients and water due to their size and less developed roots system. This resource limitation can influence the production and diversity of chemical compounds. In response to resource scarcity, plants may prioritize the production of a smaller number of compounds essential to their survival, resulting in low chemodiversity in young plants [77]. Another point is that smaller plants may be at earlier stages of development, in which they invest more energy in growth and vegetative expansion than in the production of secondary compounds. As plants grow and mature, they can allocate more resources to producing a wider variety of chemical compounds. However, the low richness of compounds in smaller plants may rely on specific defense strategies, such as producing a single chemical compound, such as apiol, to protect them against herbivores or pathogens. As plants grow and become more resistant, they can diversify their defense strategies, which leads to a greater diversity of chemical compounds [28].

Considering that leaves of *P. rivinoides* are highly attacked by herbivores in nature (unpublished data), this observed chemodiversity α in the leaves can be supported by the screening hypothesis postulated by [75]. This hypothesis suggests that chemical diversity is maintained because it increases the probability of a plant containing a potent compound or a precursor that is effective against a particular type of natural enemy, cumulatively creating a selective advantage against a diverse set of natural enemies. Thus, high chemodiversity would provide precursors of effective combinations of compounds that work synergistically against a particular type of natural enemy [90]. Although the diversity of insects that interact with the plant has not yet been quantified (data under study), the high chemodiversity α of the leaf EO of *P. rivinoides* suggests a wide variety of herbivorous insects that attack this plant.

It is important to note that the chemodiversity α recorded for the roots of *P. rivinoides* is representative. This diversity of micromolecules may be associated with the soil microbiome, which has high biological diversity. Therefore, it can be argued that this chemodiversity may also be influenced by abiotic factors, thus being associated with the screening and the optimal defense hypothesis [3,71,75]. However, the hypothesis that plant chemical diversity influences insect diversity (niche impact theory) [10] still needs to be tested in *P. rivinoides*, since the presence and diversity of insects that interact with this species have not been quantified. Therefore, future research should investigate this relationship and thus contribute to a better understanding of the factors that influence biodiversity in terrestrial ecosystems.

The Pielou index, which infers on the evenness of compounds in the leaf EO, reached a value closer to 1; however, it is not enough to affirm that there was no dominance of one or a few compounds, representing an average uniformity. The roots EO was the least uniform among all samples of EO, because of the predominant presence of apiol, which represents more than 59% of the mixture.

The results of the analysis of chemodiversity β showed that the roots and leaves of *P. rivinoides* are chemically more similar, not due to the difference in their major constituents, but rather due to the greater diversity of chemical compounds present in these organs. In other words, the EO of the leaves are more like the EO of the roots. As mentioned earlier, leaves and roots are more susceptible to the effects of stress from their surrounding environment. However, chemodiversity β demonstrated a dissimilarity between all samples, including those from different organs and ontogeny, except for Phase IV and V where *P. rivinoides* reached 2 and 7 m, respectively. This fact is more understandable, as the plant must have reached maturity and the biochemical processes leading to the formation of secondary metabolites are well established. To date, there is no literature data on chemodiversity β for Piperaceae EO, making any comparison impossible.

The Ramos and Moreira index (EI_R&M_) for complex mixtures allows for the evaluation of the molecular oxidation–reduction patterns of a mixture, serving as a suitable tool to understand the relationship of plant biosynthesis on a fluid temporal scale, based on a non-static model. Through this index, one can infer whether a mixture is more oxidized (value obtained above zero) or more reduced (value obtained below zero) [39]. The results presented here indicate that the roots and leaves are organs with higher accumulation of oxidized compounds. The different functions performed by the plant parts reveal distinct enzymatic expressions and, consequently, reflect the value of R&M of the mixtures. It is important to consider that leaves are responsible for gas exchange and photosynthesis; therefore, they produce more reactive oxygen species (ROS), and nitrogen species (RNS) can be formed. Thus, chemical compounds from leaves may scavenge these oxidant species that can lead damages, for example, to the cell wall or DNA [39]. Conversely, the stems present the most reduced mixture, a result of low local biosynthetic activity. In relation to ontogeny, the EI_R&M_ index did not show any variation (from −3.0 to −4.0), However, the different organs also showed negative values, indicating reduced volatile mixtures. From phase I to phase V, it can be supposed that redox homeostasis ensures plant fitness.

The application of Chemodiversity and advanced Chemophenetic indices represents a relatively recent approach, which makes the study of chemical phenotypic variation even more important [39]. Our group has addressed this issue, producing a scientific review article for dissemination of these indices, and demonstrating their importance [39].

## 5. Conclusions

In summary, this study provides a comprehensive analysis of the complex dynamics of chemical diversity in *P. rivinoides* across its various developmental stages and different organs. The results offer innovative insights into the ecological and physiological mechanisms underlying the adaptive strategies of this Piperaceae species. The observed dichotomy in chemical composition (mevalonate vs. shikimate pathways), along with the consistent presence of α-pinene in all organs and developmental stages, suggests the existence of a characteristic metabolic signature in *P. rivinoides*. This signature is likely an eco-physiological response aimed at maximizing the plant’s fitness and survival in its native environment. The findings suggest that *P. rivinoides* may have developed this metabolic strategy as an adaptive mechanism, although further research is needed to confirm whether this response is indeed a targeted adaptation or a more generalized survival strategy. The concept of ontogenetic chemical niche transition, widely employed in population and community ecology, is introduced here as a framework that can also be applied to chemical ecology studies.

Furthermore, the findings of this study emphasize the importance of accurately assessing plant chemical diversity to understand fundamental aspects such as functional traits, eco-physiology, and ecological interactions within complex ecosystems. By deciphering the specific chemical signatures of *P. rivinoides*, this research provides a solid foundation for conservation and sustainable management initiatives in natural habitats where the species is found.

The phenotypic plasticity observed in the chemical composition of *P. rivinoides* essential oils underscores its ability to adapt to a wide range of environmental conditions, highlighting its relevance not only for biodiversity conservation but also for providing ecosystem services and practical applications such as ecological crop management. To further advance our understanding of the factors influencing the phenotypic variation of *P. rivinoides*’ EO and explore the full potential of this species, continuous eco-physiological investigations are needed. This is a crucial step in deepening our knowledge of plant biology and developing effective ecosystem management strategies in a globally changing scenario.

## Figures and Tables

**Figure 1 plants-13-02599-f001:**
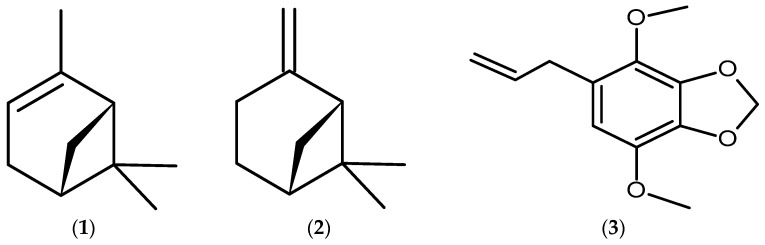
Chemical structure of the major compounds identified in the essential oils obtained from different plant organs of *P. rivinoides* Kunth. Legend: (**1**) α-pinene, (**2**) β-pinene, (**3**) apiole.

**Figure 2 plants-13-02599-f002:**
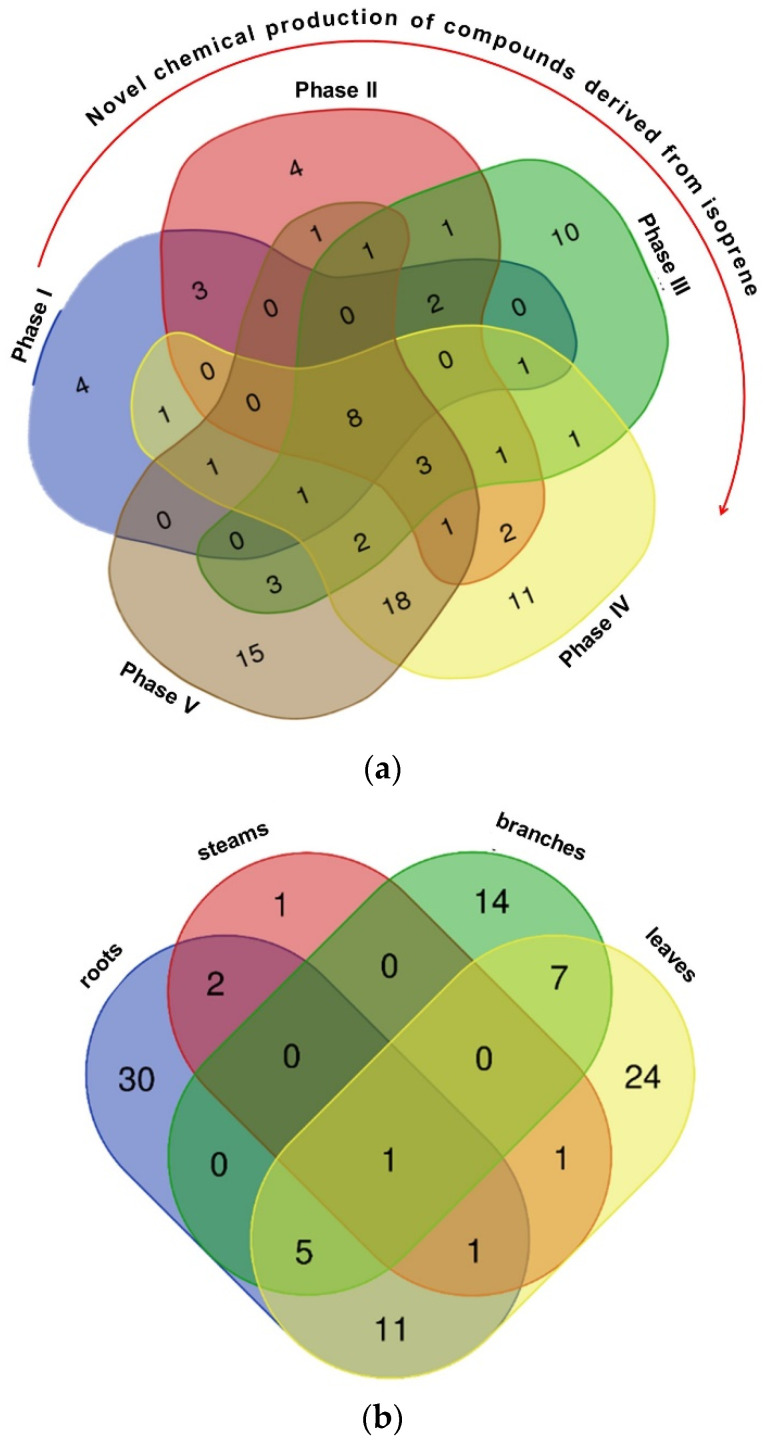
Venn diagram representing the compounds identified in the essential oil of *P. rivinoides* at (**a**) different ontogenetic stages, and (**b**) in different plant organs.

**Figure 3 plants-13-02599-f003:**
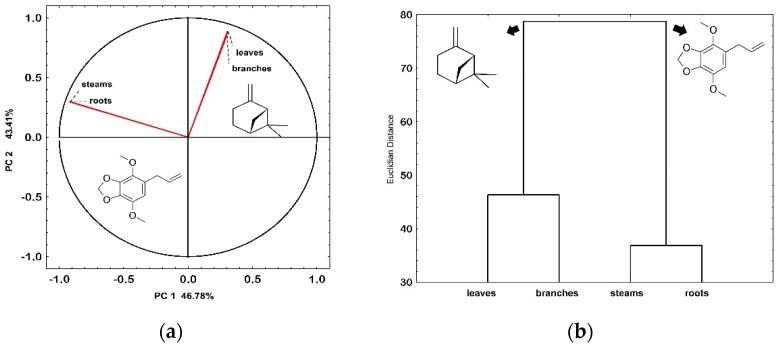
Principal Component Analysis (**a**) and Dendrogram (**b**) based on the compounds by plant organ (roots, stems, branches, and leaves) obtained from the essential oil of *P. rivinoides* Kunth.

**Figure 4 plants-13-02599-f004:**
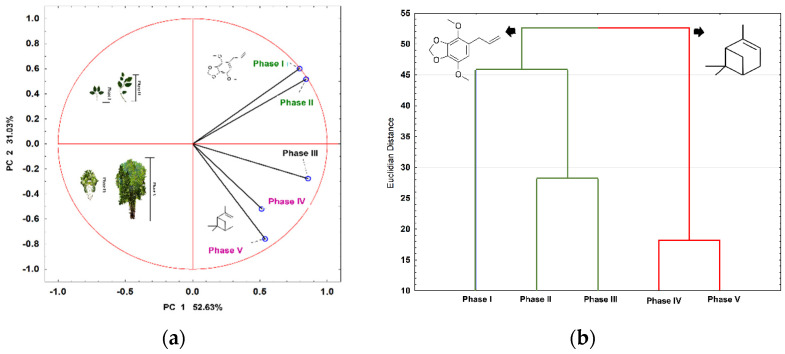
Principal Component Analysis (**a**) and Dendrogram (**b**) based on the compounds by ontogeny plant obtained from the essential oil of *P. rivinoides*.

**Figure 5 plants-13-02599-f005:**
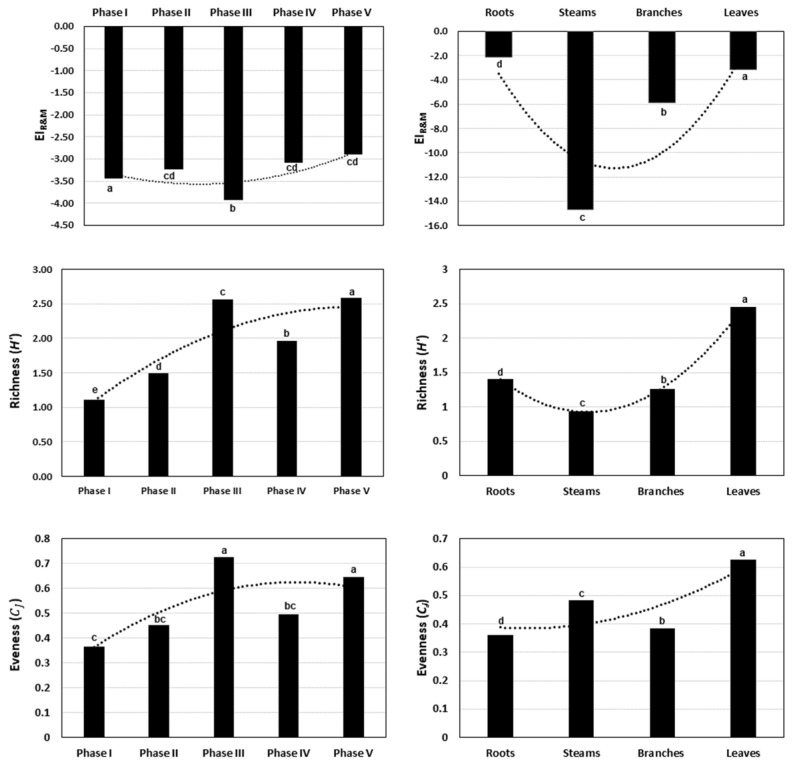
Variation among the calculated values for the Ramos and Moreira (EI_R&M_), Shannon (richness), and Pielou (evenness) indices for ontogeny and for different plant organs. The same letters mean no statistical difference (*p* > 0.05).

**Table 1 plants-13-02599-t001:** Chemical constitution and data on the essential oils of different organs of *P. rivinoides* (vegetative phenological stage) from Pedra Branca State Park/RJ.

Class	Compounds ^#^	RI_lit_	RI_cal_	Molecular Formula	Relative Percentage (Mean ± Standard Deviation) *			
					Roots	Steams	Branches	Leaves
NO	Santoline triene	906	906	C_10_H_16_	-	-	0.15 ± 0.10	-
NO	α-Thujene	924	924	C_10_H_16_	2.68 ± 0.03	-	-	0.47 ± 0.03
NO	**α-Pinene**	**932**	933	C_10_H_16_	1.07 ± 0.05	1.47 ± 0.06	**20.87 ± 0.21**	**31.90 ± 0.44**
NO	Camphene	946	945	C_10_H_16_	0.04 ± 0.02	-	0.24 ± 0.40	1.68 ± 0.22
NO	Sabinene	969	969	C_10_H_16_	0.51 ± 0.03	-	-	0.37 ± 0.03
NO	**β-Pinene**	**974**	973	C_10_H_16_	0.23 ± 0.01	-	**64.61 ± 0.22**	**20.96 ± 0.47**
NO	Myrcene	988	987	C_10_H_16_	0.43 ± 0.02	-	-	3.23 ± 0.50
NO	α-Phelandrene	1002	1000	C_10_H_16_	0.12 ± 0.04	-	-	0.63 ± 0.18
NO	α-Terpinene	1014	1012	C_10_H_16_	-	-	0.03 ± 0.04	0.24 ± 0.17
NO	*p*-Cymene	1020	1020	C_10_H_16_	0.64 ± 0.03	-	-	0.35 ± 0.61
NO	Limonene	1024	1022	C_10_H_16_	0.59 ± 0.03	-	3.76 ± 0.45	1.80 ± 0.19
NO	β-Phellandrene	1025	1024	C_10_H_16_	0.13 ± 0.07	-	-	1.06 ± 0.05
NO	*Z*-β-Ocimene	1032	1034	C_10_H_16_	-	-	0.29 ± 0.04	-
NO	*E-*β-Ocimene	1044	1048	C_10_H_16_	-	-	0.42 ± 0.02	0.20 ± 0.10
NO	γ-Terpinene	1054	1055	C_10_H_16_	0.1 ± 0.01	-	0.52 ± 0.03	0.35 ± 0.02
OM	*Z*-Sabinene hydrate	1065	1068	C_10_H_18_O	-	-	0.99 ± 0.22	0.08 ± 0.01
OT	3-Isopropyl-2-methoxypyrazine	1090	1094	C_8_H_12_N_2_O	0.05 ± 0.01	-	-	-
OM	Linalool	1095	1096	C_10_H_22_O	-	-	1.73 ± 0.05	1.83 ± 0.04
OM	*E*-Sabinene hydrate	1098	1097	C_12_H_20_O_2_	-	-	-	0.24 ± 0.01
OM	iso-3-Turjanol	1134	1132	C_10_H_18_O	0.04 ± 0.01	-	-	-
OM	*Z*-*p*-Menth-2-en-ol	1136	1136	C_10_H_18_O	-	-	-	0.14 ± 0.01
OM	Camphor	1141	1142	C_10_H_16_O	0.76 ± 0.02	-	-	-
OM	α-Terpineol	1186	1187	C_10_H_18_O	-	-	0.93 ± 0.13	-
OM	Verbenone	1204	1208	C_10_H_14_O	-	-	0.12 ± 0.07	-
OM	Piperitone	1249	1251	C_10_H_16_O	0.17 ± 0.04	-	-	-
AR	*E*-Anetole	1282	1285	C_10_H_12_O	-	-	0.53 ± 0.04	0.15 ± 0.34
AR	Safrole	1285	1289	C_10_H_10_O_2_	0.08 ± 0.01	-	-	-
OM	*Z*-Sabinyl acetate	1289	1294	C_12_H_18_O_2_	-	-	0.12 ± 0.02	-
OM	Tujanol acetate	1295	1298	C_12_H_20_O_2_	-	-	0.44 ± 0.03	-
NS	δ-Elemene	1335	1339	C_15_H_24_	-	-	0.33 ± 0.01	0.14 ± 0.17
OS	α-Terpinyl acetate	1346	1348	C_12_H_20_O_2_	-	-	-	1.54 ± 0.04
OS	Neryl acetate	1359	1362	C_12_H_20_O_2_	-	-	0.21 ± 0.04	0.35 ± 0.02
NS	α-Copaene	1374	1375	C_15_H_24_	-	-	-	0.40 ± 0.22
NS	Isoledene	1374	1378	C_15_H_25_	1.02 ± 0.03	-	-	1.48 ± 0.32
NS	α-Cubebene	1376	1380	C15H_26_	0.17 ± 0.02	-	-	-
OS	Mirtanol acetate	1385	1386	C_12_H_20_O_2_	-	-	0.26 ± 0.05	-
NS	β-Cubebene	1387	1392	C_15_H_24_	0.05 ± 0.01	-	-	-
NS	Ciperene	1398	1403	C_15_H_24_	-	-	-	0.21 ± 0.21
NS	Sibirene	1400	1409	C_15_H_24_	0.10 ± 0.01	-	-	-
NS	*E*-Caryophyllene	1417	1426	C_15_H_24_	0.13 ± 0.03	-	1.30 ± 0.04	3.03 ± 0.24
NS	β-Copaene	1430	1448	C_15_H_24_	0.04 ± 0.02	-	-	-
NS	β-Gurjunene	1433	1450	C_15_H_24_	0.28 ± 0.01	-	-	-
NS	Aromadendrene	1439	1452	C_15_H_24_	-	-	0.28 ± 0.02	2.41 ± 0.18
NS	Myltayl-4(12)-ene	1445	1444	C_15_H_24_	-	-	-	0.16 ± 0.02
NS	Muurola-3,5-diene	1448	1452	C_15_H_24_	-	-	0.10 ± 0.01	1.31 ± 0.13
NS	α-Humulene	1452	1454	C_15_H_24_	-	-	-	0.24 ± 0.02
AR	Croweacin	1457	1563	C_11_H_12_O_3_	0.05 ± 0.01	-	-	-
NS	*allo*-Aromadendrene	1458	1465	C_15_H_24_	-	-	0.17 ± 0.03	-
NS	*Z*-Cadina-1(6),4-diene	1461	1467	C_15_H_24_	-	-	-	0.28 ± 0.03
NS	γ-Gurjunene	1475	1478	C_15_H_24_	0.27 ± 0.02	-	-	-
NS	γ-Muurolene	1479	1481	C_15_H_24_	0.25 ± 0.02	-	-	-
NS	Germacrene D	1480	1490	C_15_H_24_	0.10 ± 0.01	-	-	-
NS	*E*-Muurola-4(14),5-diene	1493	1492	C_15_H_24_	0.26 ± 0.01	-	-	-
OS	*epi*-Cubebol	1493	1499	C_15_H_26_O	1.06 ± 0.04	-	-	1.33 ± 0.32
NS	Bicyclogermacrene	1500	1507	C_15_H_24_	-	-	0.23 ± 0.02	5.90 ± 0.68
NS	*E-*β-Guaienol	1502	1509	C_15_H_24_	0.46 ± 0.03	-	-	-
NS	δ-Amorphene	1511	1512	C_15_H_24_	0.46 ± 0.01	-	0.12 ± 0.01	0.10 ± 0.08
NS	γ-Cadinene	1513	1512	C_15_H_24_	**9.73 ± 0.07**	-	-	0.70 ± 0.02
OS	Cubebol	1514	1514	C_15_H_26_O	-	-	-	0.57 ± 0.06
AR	Myristicin	1517	1519	C_11_H_12_O_3_	0.05 ± 0.01	-	-	-
NS	*E*-Calamene	1521	1525	C_15_H_22_	-	-	-	4.04 ± 0.21
NS	β-Sesquifelandrene	1521	1529	C_15_H_24_	-	0.94 ± 0.12	0.42 ± 0.03	-
NS	δ-Cadinene	1522	1530	C_15_H_24_	-	-	-	0.56 ± 0.22
NS	*E*-Cadina-1,4-diene	1533	1540	C_15_H_24_	0.06 ± 0.01	-	-	-
NS	α-Cadinene	1537	1542	C_15_H_24_	-	-	-	0.56 ± 0.09
OS	Elemol	1548	1550	C_15_H_26_O	-	-	-	0.07 ± 0.01
NS	β-Calacorene	1564	1566	C_15_H_20_	0.59 ± 0.02	-	-	-
AR	Isoelemicin	1568	1569	C_12_H_16_O_3_	0.03 ± 0.01	-	-	-
AR	γ-Asarone	1572	1574	C_12_H_16_O_3_	0.05 ± 0.01	-	-	-
OS	Spathulenol	1577	1581	C_15_H_24_O	0.03 ± 0.01	-	-	2.30 ± 0.15
OS	Caryophyllene oxyde	1582	1580	C_15_H_24_O	0.17 ± 0.03	-	-	0.15 ± 0.01
OS	Globulol	1590	1585	C_15_H_26_O	-	-	0.42 ± 0.04	2.76 ± 0.55
OS	Viridiflorol	1592	1594	C_15_H_26_O	-	-	-	0.34 ± 0.02
OS	Cubeban-11-ol	1595	1596	C_15_H_26_O	-	-	-	0.17 ± 0.01
AR	6-Methoxyelemicin	1595	1599	C_13_H_18_O_4_	0.64 ± 0.03	-	-	-
OS	Rosifoliol	1600	1600	C_15_H_26_O	-	3.03 ± 0.03	-	0.41 ± 0.37
OS	Guaiol	1600	1603	C_15_H_26_O	**3.89 ± 0.06**	-	-	-
OS	1,10-di-*epi*-Cubenol	1618	1619	C_15_H_26_O	-	-	-	0.26 ± 0.10
AR	**Dillapiole**	**1620**	1628	C_12_H_14_O_4_	0.83 ± 0.03	**34.12 ± 0.46**	-	-
OS	α-Muurolol	1644	1640	C_15_H_26_O	0.17 ± 0.02	1.11 ± 0.19	-	0.94 ± 0.03
OS	Cubenol	1645	1647	C_15_H_26_O	0.86 ± 0.02	-	-	0.14 ± 0.02
OS	Agarospirol	1646	1654	C_15_H_26_O	0.12 ± 0.01	-	-	-
OS	α-Cadinol	1652	1659	C_15_H_26_O	0.04 ± 0.03	-	-	0.29 ± 0.01
OS	Selin-11-en-4-α-ol	1658	1667	C_15_H_26_O	-	-	-	-
AR	**Apiole**	**1677**	1689	C_12_H_14_O_4_	**69.68 ± 0.18**	**59.32 ± 0.24**	-	-
OS	Amorfa-4,9-dien-2-ol	1700	1718	C_15_H_24_O	0.14 ± 0.01	-	-	-
OS	5-Hydroxy-*Z*-calamenene	1713	1722	C_15_H_22_O	0.03 ± 0.01	-	-	0.11 ± 0.02
Number of compounds identified					49	6	27	50
Total Quantified Compounds					99.45	99.99	99.59	98.93
Non-Oxygenated Monoterpenes (NO)					6.54	1.47	90.89	63.24
Oxygenated Monoterpenes (OM)					0.97	0.00	4.33	2.29
Non-Oxygenated Sesquiterpenes (NS)					13.97	0.94	2.95	21.52
Oxigenated Sesquiterpenes (OS)					6.51	4.14	0.89	11.73
Arylpropanoids (AR)					71.41	93.44	0.53	0.15
Other (OT)					0.05	0.00	0.00	0.00
Yield of EO% (*w*/*w*)					0.57	0.05	0.31	0.82

Leg: **RI_cal_** = Calculated Retention Index (HP-5MS column); **RI_lit_** = Literature Retention index (Adams [32]); main constituents in bold. * Quantities are averaged out of five replicates. ^#^ All compounds were identified by MS and RI in accordance with experimental.

**Table 2 plants-13-02599-t002:** Chemical constitution and data on the essential oils of ontogeny of *P. rivinoides* (vegetative phenological stage) from the Pedra Branca State Park/RJ.

Class	Constituintes ^#^	RI_lit_	RI_cal_	Molecular Formula		Relative Percentage (Mean ± Standard Deviation) *				
						Phase I	Phase II	Phase III	Phase IV	Phase V
NO	Tricyclene	921	921	C_10_H_16_		-	-	1.2 ± 0.06	-	-
NO	Artemisiatiene	923	923	C_10_H_16_		-	-	2.03 ± 0.04	-	-
NO	α-Thujene	924	926	C_10_H_16_		-	0.02 ± 0.63	5.90 ± 0.53	0.21 ± 0.01	0.54 ± 0.06
NO	**α-Pinene**	932	933	C_10_H_16_		0.62 ± 0.66	1.68 ± 0.93	**20.88 ± 0.73**	**20.03 ± 0.14**	**35.09 ± 0.68**
NO	Camphene	946	945	C_10_H_16_		0.12 ± 0.02	0.29 ± 0.22	1.27 ± 0.13	0.70 ± 0.53	1.79 ± 0.20
NO	Sabinene	969	970	C_10_H_16_		-	-	-	1.03 ± 0.05	0.77 ± 0.20
NO	β-Pinene	974	975	C_10_H_16_		3.58 ± 0.32	7.88 ± 0.28	**11.45 ± 0.12**	**16.72 ± 0.62**	**14.48 ± 0.44**
NO	Myrcene	988	990	C_10_H_16_		0.33 ± 0.07	0.84 ± 0.04	1.54 ± 0.06	2.55 ± 0.99	3.30 ± 0.89
NO	α-Phellandrene	1002	1002	C_10_H_16_		-	0.39 ± 0.33	0.95 ± 0.23	-	-
NO	δ-2-Carene	1008	1003	C_10_H_16_		7.18 ± 0.17	16.07 ± 0.06	0.32 ± 0.03	**42.07 ± 0.78**	0.80 ± 0.60
NO	δ-3-Carene	1008	1010	C_10_H_16_		-	0.17 ± 0.92	9.91 ± 0.50	0.20 ± 1.21	0.38 ± 1.12
NO	α-Terpinene	1014	1015	C_10_H_16_		-	-	-	-	-
NO	*p*-Cymene	1020	1022	C_10_H_16_		0.12 ± 0.02	0.14 ± 0.01	0.14 ± 0.21	1.17 ± 0.01	0.58 ± 0.59
NO	Limonene	1024	1024	C_10_H_16_		0.45 ± 0.06	0.16 ± 0.01	1.55 ± 0.33	1.85 ± 0.09	1.34 ± 0.38
NO	β-Phellandrene	1025	1026	C_10_H_16_		-	-	0.10 ± 0.42	0.83 ± 0.02	1.27 ± 0.16
NO	β-Ocimene	1032	1029	C_10_H_16_		-	-	0.14 ± 0.62	-	-
NO	*Z*-β-Ocimene	1037	1033	C_10_H_16_		-	0.61 ± 0.87	-	0.04 ± 0.01	-
NO	*E*-β-Ocimene	1044	1040	C_10_H_16_		-	0.40 ± 0.71	-	0.07 ± 0.01	-
NO	γ-Terpinene	1054	1059	C_10_H_16_		-	-	-	0.20 ± 0.01	0.40 ± 0.16
OM	*Z*-Sabinenehydrate	1065	1061	C_10_H_18_O		-	-	-	0.02 ± 0.23	0.17 ± 0.25
NO	*p*-Mentha-2,4(8)-diene	1085	1080	C_10_H_16_		-	-	-	0.08 ± 0.91	0.33 ± 0.16
OM	Terpinolene	1086	1088	C_10_H_18_O		-	-	-	1.58 ± 0.11	-
NO	*p*-Cymenene	1089	1094	C_10_H_12_		-	-	-	0.16 ± 0.72	-
OM	Linalool	1095	1096	C_10_H_22_O		-	-	-	0.84 ± 0.83	1.67 ± 0.16
OM	*cis*-*p*-Menth-2-en-1-ol	1118	1112	C_10_H_18_O		-	-	-	0.04 ± 0.01	-
OM	*E*-Sabinol	1137	1134	C_10_H_16_O		-	-	-	0.03 ± 0.02	-
OM	Isopulegol	1155	1153	C_10_H_18_O		-	-	-	0.75 ± 0.01	-
OM	*p*-Mentha-1,5-dien-8-ol	1166	1163	C_10_H_16_O		-	-	-	0.05 ± 0.98	-
OM	Terpinen-4-ol	1174	1172	C_10_H_18_O		-	-	-	0.09 ± 0.42	0.34 ± 0.16
OM	α-Terpineol	1186	1188	C_10_H_18_O		0.27 ± 1.05	-	-	0.03 ± 0.10	-
OM	Piperitol	1195	1192	C_10_H_18_O		-	-	-	0.09 ± 0.32	-
OM	Pulegenol	1233	1230	C_10_H_18_O		-	-	-	0.07 ± 0.91	-
AR	*E*-Anethole	1282	1280	C_10_H_12_O		-	-	-	0.39 ± 0.42	0.45 ± 0.16
OM	α-Terpinylacetate	1346	1340	C_12_H_20_O_2_		-	-	-	-	3.19 ± 0.23
AR	Eugenol	1356	1351	C_10_H_12_O_2_		-	0.54 ± 0.63	-	-	-
NS	δ-Elemene	1335	1335	C_15_H_24_		0.18 ± 0.71	-	0.24 ± 0.82	0.07 ± 0.75	-
	NI	-	-	-		-	-	-	0.54 ± 1.91	0.11 ± 0.02
NS	α-Cubebene	1348	1346	C_15_H_24_		-	-	-	-	0.76 ± 0.10
OS	Nerylacetate	1359	1356	C_12_H_20_O_2_		-	-	-	0.08 ± 0.91	0.17 ± 0.05
NS	α-Copaene	1374	1370	C_15_H_24_		0.16 ± 0.43	-	0.32 ± 1.31	0.03 ± 0.76	0.15 ± 0.06
NS	β-Cubebene	1387	1384	C_15_H_24_		-	-	-	-	0.10 ± 0.81
NS	β-Elemene	1389	1386	C_15_H_24_		-	-	0.78 ± 0.02	-	-
NS	α-Gurjunene	1409	1416	C_15_H_24_		-	-	0.08 ± 0.01	-	0.24 ± 0.02
NS	*E*-Caryophyllene	1417	1420	C_15_H_24_		3.42 ± 0.41	3.04 ± 0.06	6.68 ± 0.12	1.53 ± 0.17	5.27 ± 0.40
NS	β-Cedrene	1419	1023	C_15_H_24_		-	0.12 ± 0.05	0.17 ± 0.71	-	0.10 ± 0.91
NS	β-Copaene	1430	1428	C_15_H_24_		0.18 ± 0.04	0.1 ± 0.91	-	-	-
NS	β-Gurjunene	1431	1429	C_15_H_24_		-	0.08 ± 0.91	-	-	0.12 ± 0.31
NS	γ-Elemene	1434	1431	C_15_H_24_		-	0.11 ± 0.83	-	-	-
NS	Aromadendrene	1439	1440	C_15_H_24_		-	-	-	0.20 ± 0.71	2.47 ± 0.34
NS	β-Barbatene	1440	1440	C_15_H_24_		-	-	-	-	0.24 ± 0.54
NS	6,9-Guaiadiene	1442	1441	C_15_H_24_		-	-	-	0.09 ± 0.99	0.50 ± 0.22
NS	α-Humulene	1452	1455	C_15_H_24_		-	-	-	0.07 ± 0.76	0.42 ± 0.98
NS	*allo*-Aromadendrene	1458	1460	C_15_H_24_		-	-	-	-	0.19 ± 0.73
NS	*cis*-Cadina-1(6),4-diene	1461	1463	C_15_H_24_			-	-	-	0.09 ± 0.62
NS	*trans*-Cadina-1(6),4-diene	1475	1477	C_15_H_24_		-	-	-	-	0.19 ± 0.81
NS	Muurola-4(14),5-diene	1465	1464	C_15_H_24_		-	-	-	0.10 ± 0.61	-
NS	Cumacrene	1470	1466	C_15_H_24_		-	0.98 ± 0.91	-	-	-
NS	γ-Gurjunene	1475	1485	C_15_H_24_		-	-	-	-	0.60 ± 0.13
NS	γ-Muurolene	1478	1486	C_15_H_24_		-	-	-	-	11.15 ± 0.28
NS	Amopha-4,7(11)-diene	1479	1489	C_15_H_24_		-	-	0.17 ± 0.21	-	0.17 ± 0.88
NS	Germacrene D	1480	1479	C_15_H_24_		-	-	2.09 ± 0.72	-	0.49 ± 0.73
NS	δ-Selinene	1492	1495	C_15_H_24_		-	-	-	-	-
NS	Bicyclogermacrene	1500	1506	C_15_H_24_		-	-	5.58 ± 0.22	0.81 ± 0.31	-
NS	δ-Amorphene	1511	1514	C_15_H_24_		-	-	-	0.09 ± 0.71	-
NS	γ-Cadinene	1513	1515	C_15_H_24_		-	-	-	0.07 ± 0.93	0.55 ± 0.87
NS	β-Curcumene	1514	1419	C_15_H_24_		-	-	-	0.26 ± 0.99	-
NS	Calamene	1521	1523	C_15_H_24_		-	-	2.28 ± 0.12	-	-
NS	*cis*-Calamenene	1528	1528	C_15_H_24_		-	-	-	0.76 ± 0.01	1.18 ± 0.03
NS	δ-Cadinene	1522	1528	C_15_H_24_		-	-	-	-	-
NS	Cadina-1,4-diene	1528	1533	C_15_H_24_		-	0.11 ± 0.22	0.09 ± 0.21	0.06 ± 0.97	-
NS	γ-Cuprenene	1532	1535	C_15_H_24_		-	-	-	-	0.29 ± 0.03
NS	Selina-3,7(11)-diene	1545	1548	C_15_H_24_		-	-	-	0.10 ± 0.21	0.19 ± 0.04
NS	*trans*-Dauca-4(11),7-diene	1557	1563	C_15_H_24_		-	-	-	-	0.52 ± 0.03
OS	Spathulenol	1577	1579	C_15_H_24_O		-	-	0.32 ± 0.41	0.77 ± 0.41	2.12 ± 0.05
OS	CaryophylleneOxide	1582	1580	C_15_H_24_O		-	-	-	0.04 ± 0.65	0.10 ± 0.02
OS	Globulol	1590	1587	C_15_H_26_O		-	0.16 ± 0.05	0.12 ± 0.21	0.95 ± 0.37	2.25 ± 0.08
OS	Carotol	1594	1598	C_15_H_26_O		-	-	-	0.07 ± 0.81	0.48 ± 0.93
OS	Rosifoliol	1600	1605	C_15_H_26_O		-	-	-	0.06 ± 0.78	0.33 ± 0.55
OS	5-*epi*-7-*epi*-α-Eudesmol	1607	1606	C_15_H_26_O		-	-		0.10 ± 0.02	0.22 ± 0.56
OS	1,10-di-*epi*-Cubenol	1618	1616	C_15_H_26_O		0.14 ± 0.02	-	-	0.35 ± 0.48	0.08 ± 0.75
OS	1-*epi*-Cubenol	1627	1631	C_15_H_26_O		-	0.31 ± 1.01	-	0.05 ± 0.91	0.10 ± 0.35
OS	*epi*-α-Muurolol	1640	1639	C_15_H_26_O		-	-	-	0.18 ± 1.21	0.55 ± 0.92
OS	α-Muurolol	1644	1640	C_15_H_26_O		-	-	-	-	0.08 ± 0.05
OS	Cubenol	1645	1641	C_15_H_26_O		-	-	-	-	0.29 ± 0.05
OS	Nerolidylacetate	1676	1679	C_17_H_28_O_2_		-	-	0.1 ± 0.04	-	-
AR	Croweacin	1457	1454	C_11_H_12_O_3_		-	0.14 ± 0.05	-	-	-
AR	Myristicin	1517	1518	C_11_H_12_O_3_		-	-	2.79 ± 0.57	-	-
AR	*E*-Carpacin	1593	1591	C_11_H_12_O_3_		0.27 ± 0.05	-	-	-	-
AR	6-Methoxy-elemicin	1595	1593	C_13_H_18_O_4_		0.70 ± 0.03	-	-	-	-
AR	Isomyristicin	1616	1614	C_11_H_12_O_3_		3.59 ± 0.17	2.69 ± 0.78	-	-	-
AR	*Z*-Asarone	1617	1615	C_12_H_16_O_3_		0.12 ± 0.04	0.31 ± 0.22	-	-	-
OT	Butylanthranilate	1617	1615	C_11_H_15_NO_2_		0.13 ± 0.92	-	-	-	-
AR	Dillapiole	1620	1618	C_12_H_14_O_4_		2.76 ± 0.17	2.29 ± 0.18	0.85 ± 0.83	-	-
AR	**Apiole**	1677	1674	C_12_H_14_O_4_		**74.69 ± 0.48**	**59.59 ± 0.56**	**15.65 ± 0.47**	-	-
AR	Niranin	1715	1713	C_11_H_15_NOS		-	-	0.49 ± 0.39	-	-
OT	NI	1600	1597	C_16_H_34_		0.24 ± 0.02	-	-	-	-
OT	4-*epi*-Abietol	2343	2345	C_20_H_32_O		-	-	0.16 ± 0.51	-	-
OT	Libocedrol	2344	2348	C_22_H_30_O_4_		-	-	0.08 ± 0.41	-	-
OT	Heyderiol	2390	2397	C_22_H_30_O_4_		-	-	0.02 ± 0.71	-	-
Number of compounds identified					20		27	34	51	54
Total Quantified Compounds					99.25		99.22	96.34	99.06	99.23
Non-Oxygenated Monoterpenes (NO)					12.40		28.65	57.38	87.91	61.07
Oxygenated Monoterpenes (OM)					0.27		0.00	0.00	3.59	5.37
Oxigenated Sesquiterpenes (OS)					0.14		4.54	18.38	4.78	25.98
Non-Oxygenated Sesquiterpenes (NS)					3.94		0.47	0.54	2.65	6.77
Arilpropanoids (AR)					82.13		65.56	19.78	0.39	0.45
Others (OT)					0.37		0.00	0.26	0.00	0.00
Yield of EO (*w*/*w*)					0.89		0.93	0.76	0.92	0.87

Leg: **RI_cal_** = Calculated Retention Index (HP-5MS column); **RI_lit_** = Literature Retention index (Adams [32]); NI: not identified; main constituents in bold. * Quantities are averaged out of five replicates. ^#^ All compounds were identified by MS and RI in accordance with experimental.

**Table 3 plants-13-02599-t003:** Calculated values for assessing chemodiversity β for the composition of essential oils obtained from different organs and for ontogenetic phases of *P. rivinoides*.

Sample Compared	Jaccard Index	Sorensen Index
roots × steams	5.77	10.91
roots × branches	10.14	18.00
roots × leaves	28.57	44.00
steams × branches	6.00	12.00
steams × leaves	4.00	7.00
branches × leaves	31.00	20.50
phase I × phase II	6.82	12.77
phase I × phase III	3.85	7.00
phase I × phase IV	18.33	31.00
phase I × phase V	15.63	27.00
phase II × phase III	36.00	52.00
phase II × phase IV	24.00	38.00
phase II × phase IV	21.00	35.00
phase III × phase IV	25.00	40.00
phase III × phase V	26.00	41.00
phase IV × phase V	75.00	86.00

## Data Availability

Data is contained within the article and Appendix A.

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
