# Peer review of "Spatio-Temporal Variations of Volatile Metabolites as an Eco-Physiological Response of a Native Species in the Tropical Forest"

_plants, 2024, doi:10.3390/plants13182599_

Round 1
Reviewer 1 Report
Comments and Suggestions for Authors
To long introduction and informative about the chemodiversity. Not many information about the plant
To long discussion
Author Response
Comments 1: To long introduction and informative about the chemodiversity. Not many information about the plant.
Respose: We would like to express our deepest gratitude for your thorough review and valuable suggestions. Your insights have been instrumental in enhancing the quality of our manuscript.
In response to your comments, we have revised the introduction by reducing its length and incorporating additional information about the plant Piper rivinoides. However, it is important to note that the available literature on this species is quite limited, with the majority of studies being conducted by our research group.
Comments 2: To long discussion
Response: Regarding the discussion, we would like to clarify that this section was carefully crafted to address not only the chemical composition of the plant but also the ecological functions of the metabolites as described in the literature. Additionally, we explored the biosynthetic pathways of these compounds, which was essential to support theoretical interpretations related to crosstalk and trade-offs. Furthermore, we provided a detailed discussion on the variations in chemodiversity and chemical diversity of the plant, correlating them with the ecological functions of substances throughout the spatiotemporal analysis conducted in the study.
Respectfully, we would like to highlight that the other reviewers found the discussion to be well-founded, with all major findings of the study being thoroughly explored. Therefore, we believe that further condensing the discussion could potentially compromise the quality and depth of the analyses presented.
Once again, we sincerely appreciate your observations and the time you have dedicated to reviewing our manuscript. Your contributions have been of great value in refining this work.
Kind regards,
Reviewer 2 Report
Comments and Suggestions for Authors
The article presents the study carried out in a clear and documented form. The articulation is good and the methods followed for the chemical and statistical analyzes are relevant. The discussion of the data is clear and effective, and the results obtained seem very interesting. The conclusions are concise but effective.
The text - apart from a few directly highlighted formal aspects - does not require further modifications.
I recommend citing article 11 bis* in line 62 and citing article 13bis** in line 67; if accepted, they should also be included in the bibliography.
The article is therefore worthy of publication on PLANTS without substantial changes. I don't express on linguistic aspects, not being English language-mother tongue!
____________________
* Cornara, L., Sgrò, F., Raimondo, F.M., Ingegnieri, M.R., Mastracci, L., D’Angelo, V., Germanò, M.P., Trombetta, D., Smeriglio, A. Pedoclimatic Conditions Influence the Morphological, Phytochemical and Biological Features of Mentha pulegium L. - Plants 2023, 12, 24. https://doi.org/10.3390/plants12010024
** Caputo, L.; Cornara, L.; Raimondo, F.M.; De Feo, V.; Vanin, S.; Denaro, M.; Trombetta, D.; Smeriglio, A. Mentha pulegium L.: A plant underestimated for its toxicity to be recovered from the perspective of the circular economy. Molecules 2021, 26, 2154.

Author Response
Comments: The article presents the study carried out in a clear and documented form. The articulation is good and the methods followed for the chemical and statistical analyzes are relevant. The discussion of the data is clear and effective, and the results obtained seem very interesting. The conclusions are concise but effective.
The text - apart from a few directly highlighted formal aspects - does not require further modifications.
I recommend citing article 11 bis* in line 62 and citing article 13bis** in line 67; if accepted, they should also be included in the bibliography.
The article is therefore worthy of publication on PLANTS without substantial changes. I don't express on linguistic aspects, not being English language-mother tongue!
Responses: Dear Reviewer,
We sincerely appreciate your thorough review and the positive feedback on our work. Your insights and comments have been very encouraging.
Regarding your suggestion to include additional references, we have carefully evaluated them. We agree that the inclusion of the reference https://doi.org/10.3390/plants12010024 is indeed valuable. This reference contributes significantly to the discussion on the influence of abiotic factors on the chemical phenotypic plasticity of plants, and we are grateful for your suggestion.
However, regarding the second suggested article, while it also appears to be relevant, our manuscript already includes similar references that are equivalent in supporting our arguments. Therefore, we have opted not to include the second reference.
Once again, thank you for your constructive comments and suggestions. Your feedback has greatly contributed to the refinement of our manuscript.
Best regards,
Reviewer 3 Report
Comments and Suggestions for Authors
Some corrections to the manuscript are proposed:
1. Line 6. Write: … Lima Moreira
2. Line 8. Write: Postgraduate Program in …
3. Line 17. Check the email address for Jessica.
4. Line 22. Delete: along with leaves (as leaves had just been mentioned).
5. Line 24. Write in a no personal way (the same occurs in other parts of the manuscript, e.g. line 30,68,96,98, … 577).
6. Line 29-30. Write: ontogeny of plant parts …
7. Line 34. Piperaceae, needs to be separated from Metabolic (Revise all the manuscript because joined words is a fault that appears frequently).
8. Line 115. Delete: … and leaves of …; instead, write: … at …
9. Line 120. Write: … divided by the weight (g) of fresh plant … (and eliminate the content of parenthesis).
10. Line 165. Use only the abbreviation, as it was already defined in line 149.
11. Line 178. A separation is needed: … (1) Pi corresponds to … Rewrite the meaning of this index, as the writing is confused.
12. 180-181. Delete H’ is the value of the Shannon index (H’). Abbreviation for the Shannon index was stated at line 166.
13. Line 197. The numbers for α-pinene and β-pinene are wrong. β-Pinene has an exo double bond (By the way, numbers for compounds must be written in bold). Correct numbers in Figure 1 and verify data for both pinene compounds taking in account their wrong numbering.
14. Line 199. Number 3 must be written in bold.
15. Line 204. Use only the abbreviation PCA, as it was already defined in line 152.
16. Lines 206 and 208. According to Figure 1, apiol must be PC3.
17. Line 208. Revise data for PC1 and PC2, taking in account of their wrong numbering. Correct descriptions at Figure 3a.
18. Line 209. Use the abbreviation HCA; its meaning was already defined in line 153.
19. In Tables 1 and 2, write: 6-Methoxyelemicin (RI 1595/1599). At the end of this Table 1, a line for identified compounds shows numbers 49, 6, 27 and 50, which compounds correspond to those numbers? Are quantities below each number percentages (the same for Table 2)? At the end of Table 2 a line for identified compounds shows numbers 20, 27, 34, 51 and 54, which compounds correspond to those numbers?
20. At the end of Tables 1 and 2, the reference 10 is authored by C. Müller and R.R. Junker, not by Adams. Check the correct reference number.
21. Line 228-229. Clarify the sentence.
22. Lines 233 and 235. Written quantities at ranges for Apiole, dillapiole, α-pinene, and δ-2-carene are wrong; write just ranges quantities for phases I – II, and IV-V.
23. Line 248. The Venn diagrams are not understood as the 9 numbers at Figure 2a and the 8 numbers at Figure 2b have not a corresponding name compound. These Venn diagrams are confused.
24. Figures 3a, 3b, 4a and 4b. Check the pinene structures.
25. Line 256. Introduce examples showing how the diverse indices were calculated.
26. Line 349. Write: Pinene has two …
27. Lines 378-380. An alternative writing is suggested for this phrase.
28. Line 420. Start phrase as: Factors …
29. Lines 585-589. This part of Conclusions is not precise, conclusive; authors suggest … that … probably …
Comments on the Quality of English LanguageEnglish is good.
Author Response
Dear Reviewer,
We would like to express our deepest gratitude for your thorough review and valuable suggestions. Your insights have been instrumental in enhancing the quality of our manuscript.
In relation to your suggestions, we clarify some points below:
Line 6. Write: … Lima Moreira (DONE)
- Line 8. Write: Postgraduate Program in …(DONE)
- Line 17. Check the email address for Jessica. (DONE)
- Line 22. Delete: along with leaves (as leaves had just been mentioned). (DONE)
- Line 24. Write in a no personal way (the same occurs in other parts of the manuscript, e.g. line 30,68,96,98, … 577). (All the text was revised and any error was corrected.)
- Line 29-30. Write: ontogeny of plant parts …(DONE)
- Line 34. Piperaceae, needs to be separated from Metabolic (Revise all the manuscript because joined words is a fault that appears frequently). (DONE -All the text was revised and the error was corrected)
- Line 115. Delete: … and leaves of …; instead, write: … at …(DONE)
- Line 120. Write: … divided by the weight (g) of fresh plant … (and eliminate the content of parenthesis). (DONE)
- Line 165. Use only the abbreviation, as it was already defined in line 149. (DONE)
- Line 178. A separation is needed: … (1) Pi corresponds to … Rewrite the meaning of this index, as the writing is confused. (DONE)
- 180-181. Delete H’ is the value of the Shannon index (H’). Abbreviation for the Shannon index was stated at line 166. (DONE)
- Line 197. The numbers for α-pinene and β-pinene are wrong. β-Pinene has an exo double bond (By the way, numbers for compounds must be written in bold). Correct numbers in Figure 1 and verify data for both pinene compounds taking in account their wrong numbering. (We really sorry about this mistake, we appreciated your attention review and this error was corrected in the figure 1)
- Line 199. Number 3 must be written in bold. (DONE)
- Line 204. Use only the abbreviation PCA, as it was already defined in line 152. (DONE)
- Lines 206 and 208. According to Figure 1, apiol must be PC3. (Considering that the sum of principal components 1 and 2 is greater than 90. in this case, there is no significant need to include PC3 as a new axis.)
- Line 208. Revise data for PC1 and PC2, taking in account of their wrong numbering. Correct descriptions at Figure 3a. (DONE)
- Line 209. Use the abbreviation HCA; its meaning was already defined in line 153. (DONE)
- In Tables 1 and 2, write: 6-Methoxyelemicin (RI 1595/1599). At the end of this Table 1, a line for identified compounds shows numbers 49, 6, 27 and 50, which compounds correspond to those numbers? Are quantities below each number percentages (the same for Table 2)? At the end of Table 2 a line for identified compounds shows numbers 20, 27, 34, 51 and 54, which compounds correspond to those numbers? (DONE. The numbers represent the total number of compounds identified in each essential oil sample. The percentage at the end of the table indicates the total percentage of compounds identified in the mixture)
- At the end of Tables 1 and 2, the reference 10 is authored by C. Müller and R.R. Junker, not by Adams. Check the correct reference number. (We corrected this mistake)
- Line 228-229. Clarify the sentence. (DONE)
- Lines 233 and 235. Written quantities at ranges for Apiole, dillapiole, α-pinene, and δ-2-carene are wrong; write just ranges quantities for phases I – II, and IV-V. (DONE)
- Line 248. The Venn diagrams are not understood as the 9 numbers at Figure 2a and the 1 numbers at Figure 2b have not a corresponding name compound. These Venn diagrams are confused. (Allow me to explain the purpose of the diagram. This diagram represents a qualitative analysis aimed at showing which constituents are shared across all or some samples, thereby indicating the degree of similarity among them. In this case, the number 8 in the central area of Diagram 1 represents all the substances found in every sample; these are described in the text on lines 217-219. In the second diagram, the presence of the number 1 in the central area represents the presence of α-pinene in all the studied samples. The text has been revised for clarity and better comprehension.)
- Figures 3a, 3b, 4a and 4b. Check the pinene structures. (DONE)
- Line 256. Introduce examples showing how the diverse indices were calculated. (The articles that previously used these indices are listed in the references at positions 6 and 36-38. Additionally, the citations have been included at the end of the relevant sentence.)
- Line 349. Write: Pinene has two …(DONE)
- Lines 378-380. An alternative writing is suggested for this phrase. (The sentence has been rewritten to possibility better understood.)
- Line 420. Start phrase as: Factors …(DONE)
- Lines 585-589. This part of Conclusions is not precise, conclusive; authors suggest … that … probably …( the conclusion has been adjusted to be more accurate and conclusive.)
We would like to say thank you for your contribuition, one more time.
Round 2
Reviewer 3 Report
Comments and Suggestions for Authors
I consider that the manuscript has been improved; however, I send the following additional comments:
1. Lines 190, and 336. In this manuscript the homogeneous name for apiole contains an end e.
2. Lines 223-224. The correct range for apiole is 74.69-59.59 and for dillapiole 2.76-2.29.
3. Figure 2 shows Venn diagrams that exhibit how many compounds were found in Phases I-V, and in 4 plant organs; however, more important is to know their compound names. In lines 218-219 only those compounds present in all stages of ontogeny are described.
4. Line 339. Instead of … has two…; write: … is a common …
Author Response
[comments] I consider that the manuscript has been improved; however, I send the following additional comments:
- Lines 190, and 336. In this manuscript the homogeneous name for apiole contains an end e.
- Lines 223-224. The correct range for apiole is 74.69-59.59 and for dillapiole 2.76-2.29.
- Figure 2 shows Venn diagrams that exhibit how many compounds were found in Phases I-V, and in 4 plant organs; however, more important is to know their compound names. In lines 218-219 only those compounds present in all stages of ontogeny are described.
- Line 339. Instead of … has two…; write: … is a common
[response]
Dear Reviewer,
We greatly appreciate the level of review, as well as the care and attention dedicated to our work.
1- DONE
2 - DONE
3- Thank you for your observation regarding the inclusion of the names of the substances that occur simultaneously in each intersection of the Venn diagram presented in Figure 2. I understand your suggestion to describe these substances in the main text of the manuscript. However, I am concerned that adding this information to the main text could make the reading somewhat redundant, as the substances that occur simultaneously in each organ or phase are already detailed in Tables 1 and 2, which provide a clear and direct view of the percentage contents and their comparisons.
Moreover, other reviewers have considered the article quite extensive and requested the reduction of information that was not objective or sufficiently clear. Although we have a table file generated by the Venn diagram with the description of the substances that occur simultaneously in each organ, I believe it is not necessary to include all the names again, since Tables 1 and 2 were carefully constructed to enable an efficient comparison between the different results obtained.
However, in accordance with your request, we are including the Venn diagram table as supplementary material. We appreciate your understanding and the valuable contribution to the manuscript's review.
4- DONE